# Correction of estimates of retention in care among a cohort of HIV-positive patients in Uganda in the period before starting ART: a sampling-based approach

Patience Nyakato,[1,2] Agnes N Kiragga,[2] Andrew Kambugu,[2] John Bradley,[1] Kathy Baisley[1]

JB and KB are joint senior authors on this work

[1]MRC Tropical Epidemiology Group, Department of Infectious Disease Epidemiology, London School of Hygiene and Tropical Medicine, London, UK
[2]Infectious Diseases Institute, College of Health Sciences, Makerere University, Kampala, Uganda

**Correspondence to**
Miss Patience Nyakato;
pnyakato2@gmail.com

## ABSTRACT

**Objective** The aim of this study was to use a sampling-based approach to obtain estimates of retention in HIV care before initiation of antiretroviral treatment (ART), corrected for outcomes in patients who were lost according to clinic registers.

**Design** Retrospective cohort study of HIV-positive individuals not yet eligible for ART (CD4 >500).

**Setting** Three urban and three rural HIV care clinics in Uganda; information was extracted from the clinic registers for all patients who had registered for pre-ART care between January and August 2015.

**Participants** A random sample of patients who were lost according to the clinic registers (>3 months late to scheduled visit) was traced to ascertain their outcomes.

**Outcome measures** The proportion of patients lost from care was estimated using a competing risks approach, first based on the information in the clinic records alone and then using inverse probability weights to incorporate the results from tracing. Cox regression was used to determine factors associated with loss from care.

**Results** Of 1153 patients registered for pre-ART care (68% women, median age 29 years, median CD4 count 645 cells/µL), 307 (27%) were lost according to clinic records. Among these, 195 (63%) were selected for tracing; outcomes were ascertained in 118 (61%). Seven patients (6%) had died, 40 (34%) were in care elsewhere and 71 (60%) were out of care. Loss from care at 9 months was 30.2% (95% CI 27.3% to 33.5%). After incorporating outcomes from tracing, loss from care decreased to 18.5% (95% CI 13.8% to 23.6%).

**Conclusion** Estimates of loss from HIV care may be too high if based on routine clinic data alone. A sampling-based approach is a feasible way of obtaining more accurate estimates of retention, accounting for transfers to other clinics.

## INTRODUCTION

Access to antiretroviral therapy (ART) has expanded considerably in sub-Saharan Africa (SSA), with 12 million people in the region receiving ART in 2016.[1] With the UNAIDS

### Strengths and limitations of this study

► Most studies that use tracing to estimate retention in care focus on HIV-positive individuals on antiretroviral treatment (ART); this is one of few studies to apply these methods in the period before ART initiation.
► A sampling-based approach is feasible and provides an opportunity to obtain more accurate estimates of retention in HIV care programmes in resource-limited settings.
► Outcomes were not ascertained in all patients who were traced, so individuals who were traced successfully may not be representative of all who were lost.
► The follow-up time was relatively short, so some patients who were considered lost according to the clinic registers may have returned to the clinic at a later date.

90-90-90 targets (90% of HIV-positive individuals know their status, 90% of those diagnosed are on ART and 90% of those on ART are virally suppressed by 2020), more HIV-positive individuals are expected to be on ART and attain viral suppression.[2–4] However, although treatment coverage in SSA doubled between 2010 and 2015, estimated coverage was still only 47% in 2015, and HIV-related mortality remains high, partly due to loss from care.[1 2]

Major gaps in HIV treatment programmes include linking individuals who test HIV positive to care and prompt initiation of ART. Previous WHO treatment guidelines were based on CD4 count thresholds, with pre-ART care focused on immunological monitoring until individuals became eligible for ART. A 2012 systematic review of retention in HIV care in SSA found a median of 57% of individuals returned for CD4 count results after

testing HIV positive, and among those who received their results, 45% remained in care until they became eligible for ART.[5] Even among those who were ART eligible, only a median of 66% initiated ART.[5]

WHO released new HIV treatment guidelines in 2015, recommending that ART be offered to all HIV-positive individuals irrespective of CD4 count.[6] If widely implemented, the 'treat all' approach would contribute significantly to achieving the 90-90-90 goals. As of end 2016, many countries in SSA had begun implementing the new guidelines. However, several countries had introduced the guidelines only in selected treatment sites, and others had not yet adopted the new policy.[1] Scale-up of ART treatment for all HIV-positive people in resource-limited settings will require broad health systems strengthening, which in practice may mean that, for budgetary or other practical reasons, some clinics may still use CD4 counts to prioritise starting treatment. Uganda officially rolled out the test and treat guidelines in November 2016. By end 2017, nearly all government clinics had implemented test and treat. However, in practice, priority for ART initiation is given to existing patients in pre-ART care. Furthermore, ART 'stock outs' are still common, so individuals who are newly diagnosed are likely to have some period of pre-ART care. In March 2017, an estimated 6% of HIV-positive persons who were enrolled in HIV care were not on ART.[7] In addition, 48% of men who had tested HIV positive have not yet initiated ART.[7]

Under the 'treat all' guidelines, many individuals who are entering HIV care will have high CD4 counts and be asymptomatic, and therefore face different barriers to starting ART. Losses between testing HIV positive and ART initiation are still likely to remain. Obtaining accurate estimates of loss from care and outcomes in this stage will be important for evaluating the impact of the 'treat all' guidelines, and for designing interventions to improve retention and increase the numbers starting ART.[8]

Standard estimates of retention consider those who are lost to represent disengagement from care. However, many of those lost may include transfers to other care centres.[9 10] In rural Uganda, estimates of patient retention 3 years after initiating ART increased from 60% to 85% when corrected for outcomes among those who were lost.[9 11] Therefore, estimates of retention that consider patients who are lost from care to have disengaged from care are biased, and may result in misdirection of resources at the clinic and national levels.

We used a sampling-based approach to obtain more accurate estimates of loss from pre-ART care among HIV-positive individuals attending clinics in Uganda who were not yet eligible for ART. This approach involves tracing a sample of lost patients and using a weighted analysis to correct the estimates of retention in the entire clinic population, with the assumption that the traced patients are representative of all who were lost.[12–14] We also explore the effect of this approach on estimates of mortality and factors associated with loss from care before ART initiation.

## METHODS
### Study setting
This was a retrospective cohort study of patients who had registered in pre-ART HIV care at six government clinics in Uganda: three urban Kampala city municipal clinics (Kisenyi, Kawala and Kitebi), which are managed by the Kampala City Council Authority (KCCA), and three rural health centres in Hoima and Kibaale districts in western Uganda, which are run by the Uganda Ministry of Health. The three urban clinics are level IV health centres, which have a target catchment population of 100 000. Kisenyi is the largest of these and serves a particularly economically impoverished area. The rural facilities included one level IV (Kiogorobya) and one level III (Dwoli) health centre in Hoima district, and Kagadi District Hospital in Kibaale district. Hoima district had an estimated population of 574 000 in the 2014 census, and Kibaale had an estimated population of 789 000.

All six facilities are supported by the Infectious Diseases Institute (IDI), Makerere University and offer integrated HIV testing and care facilities, provision of ART and laboratory support. Most persons attend the HIV counselling and testing facilities as walk-ins. Individuals who test positive are referred for care within the clinic, or a preferred alternate facility, for immediate CD4 testing. Individuals are registered at the clinic at the time of initiating CD4 testing. They are then asked to return to the clinic within 2 weeks for their CD4 results. At the time of the study (2015), individuals who were not yet eligible for ART were enrolled in a general pre-ART HIV care programme, and visited the clinic every 3 months for routine clinic check-ups and cotrimoxazole prophylaxis. In 2015, the CD4 count threshold for ART initiation was ≤500 cells/µL; however, in practice, priority was given to individuals with CD4 counts <350. The three Kampala clinics are among seven KCCA clinics supported by IDI; in June 2015, there were 33 514 HIV-positive persons receiving HIV care services in these facilities, of whom 87% were on ART.[15]

### Study design
A list of all individuals who tested HIV positive between January and May 2015 (Kampala clinics) and January and August 2015 (rural clinics) and were not yet eligible for ART was obtained from the routine clinic records. Sociodemographic and routine clinical data for patients attending the Kampala clinics were extracted using the electronic patient records system (OpenMRS). For patients attending the rural clinics, the information was manually extracted from the paper-based clinic records and entered into an Access database. Patients were then classified as either (1) still in care, (2) transferred out to another clinic, (3) died or (4) lost to follow-up, based on the information available from the clinic records. Patients were counted as lost to follow-up if they were three or more months late for their last scheduled visit at the clinic, and not known to have transferred out or to have died.

From the sampling frame of all patients who were classified as lost, a random sample was selected, separately for urban clinics and rural clinics, for intensive tracing. The size of the sample was based on practical considerations of how many patients could be traced at each clinic with available resources, rather than on formal sample size calculations.

Tracing was done between November 2015 and March 2016 in both the urban and rural clinics. Tracers attempted to contact patients through phone calls and home visits, using addresses and locator forms containing secondary phone numbers, areas of residence and a map to the area of residence. For patients who could not be contacted through phone calls, home visits using locator forms were used; at least three visit attempts were made before declaring the patient unreachable.

Patients who were successfully traced were asked to provide information about their current HIV care status: whether they were registered elsewhere and were attending another clinic, whether they had started ART elsewhere and reasons for withdrawing from HIV care if not attending any HIV care facility. Patient deaths were ascertained through an interview with a close informant.

Individuals were considered to have disengaged from care if they had not registered at any other clinic and had not returned to the clinic where they were originally registered for more than 3 months after their last scheduled visit. Individuals who said they were purchasing cotrimoxazole directly from the pharmacy (ie, not under clinician's care) or they obtained drugs through relatives/friends were also considered to have disengaged from care. Individuals who reported to have registered at another clinic were asked the name of the other clinic, the date of their next appointment and for evidence of registration such as a patient card with a current appointment date. Those who had a current patient card, or gave a valid name of an HIV care clinic with a current appointment date, were considered to be in care.

Interviews were conducted by trained nurse counsellors, all of whom had previous research experience. Information was collected using a standard structured questionnaire. The nurse counsellors made the contact attempts by telephone and traced urban patients at their homes if the information on the locator form was sufficient. Many of the rural patients could not be reached by telephone, and the information on the locator forms was inadequate. Therefore, in the rural clinics, 'expert' patients from each clinic were paired with the nurse interviewers to help trace patients in the community. Expert patients also helped with tracing in the urban clinics, for patients who could not be found through the locator forms. Expert patients serve as community volunteers at the clinic and offer support to fellow patients in HIV care. They are familiar with the surrounding community and are often called on by the clinic to help trace persons on ART who have missed their visits.

## Statistical analysis

Descriptive statistics (proportions, means, medians and IQRs) were used to summarise baseline characteristics. Characteristics of patients who were retained in pre-ART care and those who were lost were tabulated and compared using $\chi^2$ tests, with the Rao-Scott correction to account for correlation within clinics. In addition, characteristics of patients who were selected for tracing and traced successfully were compared with those who were selected but could not be found.

First, using only the information available from the clinic records, the proportion of patients who were lost from pre-ART care was estimated using a cumulative incidence approach, where deaths known to the clinics were treated as a competing risk. Observation time began on the date of registration at the clinic (ie, the date of presenting for CD4 testing after HIV diagnosis) and ended at the earliest of the date of known transfer out, death, loss from care (defined as 3 months after last missed appointment) or review of the clinic records (for individuals who were still in care). Patients who initiated ART were censored on the date of ART initiation. Then, a corrected analysis was conducted using the same approach but incorporating the outcomes obtained from tracing. The outcomes of patients who were successfully traced were weighted using inverse probability weights, calculated as the inverse proportion of patients who were successfully traced among all patients who were lost. Patients who could not be found, or who were not selected for tracing, were given a weight of 0. Patients who were still in care according to the clinic registers were given a weight of 1. Weights were calculated separately for each clinic. For example, suppose that in a clinic with 100 patients of whom 30 were lost, 10 were successfully traced and 6 were found to be still in care, the weights for the patients who were traced would be 30/10=3. The corrected estimate for the proportion in care would be calculated as: (70×1+6 (found to be still in care)×3)/ (70×1+6×3+4 (found to be out of care)×3)=88/100. For individuals who were traced and found to still be in care, observation time was considered to end at the date of interview. Individuals who were traced and found to be alive but not in care were considered to have been lost 3 months after their last missed appointment at the original clinic.

A sensitivity analysis was also done: first, we assumed that all individuals who were traced and not found were alive and in care elsewhere; second, we assumed that all patients who were not found were alive but not in care. CIs for the weighted estimates were obtained through bootstrapping using percentiles of the bootstrap distribution with 2000 replications.

Cox proportional hazards models were fitted to examine factors associated with loss from care, using data from the clinic registers alone and in a weighted analysis after incorporating results from tracing. Robust SEs were used to account for correlation within clinics. Owing to the small number of covariates available, all variables

were included in the final multivariate model. In the rural clinics, data on clinical covariates were often missing from the patient records; therefore, the analysis of clinical covariates was restricted to patients from the urban clinics. The appropriate functional forms of continuous covariates were explored using low order polynomials (quadratic and cubic forms). All analyses were done using Stata V.14.2 (Stata Corp, College Station, Texas, USA).

### Ethics

Patients give informed consent at the time of registration in HIV care at the clinics, to be traced in case they miss their appointments. Patients who were traced successfully gave additional written or oral (phone interviews) informed consent for participation in the current study.

## RESULTS

Between the period of January and August 2015, 1153 individuals had registered in pre-ART care at the six clinics: 925 (80.2%) at the urban clinics and 228 (19.8%) at the rural clinics. A total of 307 (26.6%) individuals were classified as lost from care (table 1); 207 from the urban clinics (22.4% of urban patients) and 100 from the rural clinics (43.9% of rural patients). A random sample of 195 (63.5% of those lost) patients was selected for tracing (116 from the urban clinics and 79 from the rural clinics) and 118 (60.5%) were successfully traced. Seventy patients had face-to-face interviews in the clinics, 20 had telephone interviews and 28 had home visits.

The median (IQR) age of all patients who registered in pre-ART care was 29 (24–35) years; the majority (68.2%) were women, and the median (IQR) CD4 count was 645 (529–834) cells/µL. CD4 counts were missing for 15% of patients (10% of those still in care and 27% of those who were lost); all missing data were from the rural clinics. Characteristics of patients who were still in care were generally similar to those who were lost, but there was some evidence that those who were lost were most likely to be from rural clinics and to have higher CD4 counts (table 1). Among the 195 patients who were selected for tracing, there was no evidence of a difference in the characteristics between those who were successfully traced and those who were not found. Of those who were successfully traced, 40 (33.9%) were found to be actively in care (ie, had re-registered at another clinic and were keeping up with their clinic appointments) and 71 (60.2%) were out of care. Seven (5.9%) individuals were found to have died after having left care.

At 9 months, the cumulative incidence of loss from care based on the clinic registers was 30.2% (95% CI 27.3% to 33.5%; figure 1). After incorporating outcomes from those who were successfully traced, loss from care reduced to 18.5% (95% CI 13.8% to 23.6%). From the sensitivity analysis, assuming that the individuals who were traced but not found were all in care, then loss was 14.9% (95% CI 10.8% to 19.6%). Assuming that these patients were all out of care, then loss from care increased to 38.5% (95% CI 31.5% to 45.7%). Loss from care was higher in rural than urban clinics (46.1% vs 25.8%, respectively, based on the clinic registers). When corrected for the outcomes of those who were traced, loss from care was 28.8% (95% CI 19.9% to 37.5%) in the rural clinics and 15.3% (95% CI 9.9% to 21.5%) in the urban clinics.

Based on the information available from the clinic registers alone, no patients were known to have died. After tracing, seven patients were found to have died. After incorporating the deaths that were found through tracing, the cumulative incidence of mortality at 9 months was estimated to be 1.6% (95% CI 0.5% to 3.0%).

In both the uncorrected and corrected analysis of factors associated with loss from care, there was strong evidence that patients from rural clinics were more likely to be lost from care than those from urban clinics (adjusted(a)HR (uncorrected)=1.95, 95% CI 1.68 to 2.27, p<0.001; aHR (corrected)=2.02, 95% CI 1.49 to 2.73, p<0.001; table 2). There was some evidence that older patients were less likely to be lost from care than younger patients (aHR (uncorrected)=0.79 for each 10-year increase in age, 95% CI 0.66 to 0.94, p=0.007; aHR (corrected)=0.71; 95% CI 0.54 to 0.93, p=0.01). In the corrected analysis, but not in the uncorrected, there was also weak evidence that men were more likely to be lost from care.

Among patients from the urban clinics, in the uncorrected analysis, weight at registration was the only clinical characteristic associated with loss from care. Loss from care decreased with increasing weight to around 60 kg, and then increased. After incorporating outcomes from the successfully tracked patients, weight at registration was still associated with loss from care, but the direction of the association had changed, with the risk of loss from care increasing slightly with increasing weight to around 60 kg, and then decreasing (table 2).

Among the 71 patients who were successfully tracked and found not to be seeking care elsewhere, the main reasons for stopping care were that they lacked money for transport (37%), that they did not feel unwell (27%) or that they had moved to places without an HIV care facility (27%) (table 3). Patients also reported that they lacked time (15%), purchased cotrimoxazole from other sources (14%) or did not believe that they were HIV positive (11%). The main reasons for stopping care among urban patients was not feeling unwell (41%) or having moved (39%). Among rural patients, the main reasons were lack of money for transport (50%) or that the clinic was too far away (43%).

Among the 40 patients who reported being in care at another clinic, the main reasons for changing clinics was that the new clinic was closer to work or home (45%), they lacked money for transport (25%) or the new clinics had less waiting time (17%). The new clinic being closer was cited as the main reason for changing for both urban and rural patients (52% and 35%, respectively).

**Table 1** Characteristics of patients registered for pre-ART care

| Characteristics | In care (n=864) N (col %) | Lost (n=307) N (col %) | P values* | Tracked (n=195) N (col %) | Not tracked (n=112) N (col %) | P values† | Found (n=118) N (col %) | Not found (n=77) N (col %) | P values‡ |
|---|---|---|---|---|---|---|---|---|---|
| Sex | | | 0.17 | | | 0.97 | | | 0.84 |
| Male | 258 (30.5%) | 109 (35.5%) | | 69 (35.4%) | 72 (64.3%) | | 41 (34.7%) | 28 (36.4%) | |
| Female | 588 (69.5%) | 198 (64.5%) | | 126 (64.6%) | 40 (35.7%) | | 77 (65.3%) | 49 (63.6%) | |
| Location | | | 0.02 | | | 0.13 | | | 0.38 |
| Urban sites | 718 (84.9%) | 207 (67.4%) | | 116 (59.5%) | 91 (81.3%) | | 66 (55.9%) | 50 (64.9%) | |
| Rural sites | 128 (15.1%) | 100 (32.6%) | | 79 (40.5%) | 21 (18.8%) | | 52 (44.1%) | 27 (35.1%) | |
| Age in years | | | 0.15 | | | 0.25 | | | 0.44 |
| <20 | 27 (3.2 %) | 18 (5.9 %) | | 14 (7.3 %) | 4 (3.6%) | | 10 (8.7 %) | 4 (5.3 %) | |
| 20–29 | 415 (49.2%) | 158 (52.1%) | | 96 (50.3%) | 62 (55.4%) | | 53 (46.1%) | 43 (56.6%) | |
| 30–39 | 266 (31.6%) | 86 (28.4%) | | 58 (30.4%) | 28 (25%) | | 35 (30.4%) | 23 (30.3%) | |
| 40–49 | 91 (10.8%) | 35 (11.6%) | | 22 (11.5%) | 13 (11.6%) | | 16 (13.9%) | 6 (7.9 %) | |
| 50+ | 44 (5.2 %) | 6 (2.0 %) | | 1 (0.5 %) | 5 (4.5%) | | 1 (0.9 %) | 0 (0.0 %) | |
| Missing | 3 | 4 | | 4 | 0 | | 3 | 1 | |
| CD4 count cell | | | 0.08 | | | 0.07 | | | 0.27 |
| 350–499 | 137 (18.0%) | 35 (15.6%) | | 21 (16.5%) | 14 (14.4%) | | 13 (17.8%) | 8 (14.8%) | |
| 500–749 | 376 (49.4%) | 100 (44.6%) | | 53 (41.7%) | 47 (48.5%) | | 34 (46.6%) | 19 (35.2%) | |
| 750+ | 248 (32.6%) | 89 (39.7%) | | 53 (41.7%) | 36 (37.1%) | | 26 (35.6%) | 27 (50.0%) | |
| Median (IQR) | 644 (525–812) | 654 (540–892) | | 659 (553–918) | 648 (534–862) | | 640 (553–885) | 734 (565–946) | |
| Missing | 85 | 83 | | 68 | 15 | | 45 | 23 | |
| Weight (kg) | | | 0.27 | | | 0.53 | | | 0.10 |
| <50 | 98 (12.5%) | 36 (15.4%) | | 19 (14.0%) | 17 (17.3%) | | 9 (11.4%) | 10 (17.5%) | |
| 50 to <60 | 285 (36.5%) | 87 (37.2%) | | 48 (35.3%) | 39 (39.8%) | | 28 (35.4%) | 20 (35.1%) | |
| 60 to <70 | 235 (30.0%) | 76 (32.5%) | | 45 (33.1%) | 31 (31.6%) | | 22 (27.8%) | 23 (40.4%) | |
| 70+ | 164 (21.0%) | 35 (15.0%) | | 24 (17.6%) | 11 (11.2%) | | 20 (25.3%) | 4 (7.0%) | |
| Median (IQR) | 60 (52–66) | 59 (52–65) | | 60 (52–65.5) | 58 (52–65) | | 60 (53–70) | 59 (51–62) | |
| Missing | 64 | 73 | | 59 | 14 | | 39 | 20 | |

*P value comparing those in care and lost, using Rao-Scott correction to $\chi^2$ test to account for clustered sampling. Individuals with missing values excluded from comparison.
†P value comparing those selected for tracing and not selected, calculated as described in footnote *.
‡P value comparing those successfully traced and those not found, calculated as described in footnote *.
ART, antiretroviral treatment.

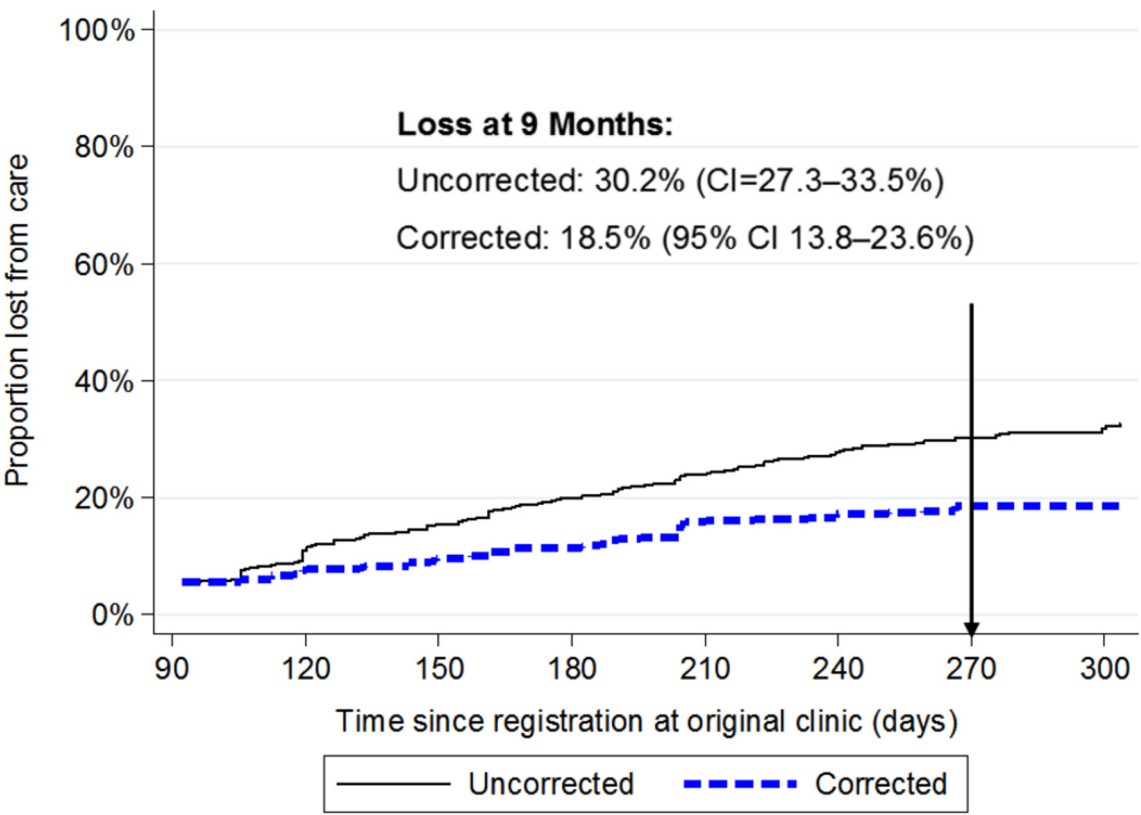

**Figure 1** Uncorrected and corrected cumulative incidence of loss from care among patients with CD4 >500 registered for HIV care at six clinics.

## DISCUSSION

Much of the research that has been done regarding correction of estimates of retention in HIV care has concentrated on HIV-positive individuals on ART. This study looked at individuals who had recently received an HIV diagnosis but were not yet eligible for ART. Based on the information

| Characteristics | Uncorrected HR* (95% CI) | P values | Corrected HR* (95% CI) | P values |
|---|---|---|---|---|
| **Table 2** Factors associated with loss from care, estimated from Cox proportional hazards models, based on data in clinic registers (uncorrected) and corrected for the outcomes among patients who were successfully traced | | | | |
| Sociodemographic | | | | |
| Sex | | 0.38 | | 0.07 |
| Female | 1 | | | |
| Male | 1.17 (0.82 to 1.68) | | 1.39 (0.97 to 1.99) | |
| Age per 10 years | | 0.007 | | 0.01 |
| | 0.79 (0.66 to 0.94) | | 0.71 (0.54 to 0.93) | |
| Location | | <0.001 | | <0.001 |
| Urban | 1 | | 1 | |
| Rural | 1.95 (1.68 to 2.27) | | 2.02 (1.49 to 2.73) | |
| Clinical† | | | | |
| CD4 count (per 100 cells) | | 0.22 | | 0.41 |
| | 1.04 (0.98 to 1.10) | | 1.05 (0.93 to 1.20) | |
| Weight per 10 kg‡ | | <0.001 | | <0.001 |
| Linear term | 0.94 (0.80 to 1.11) | | 1.03 (0.74 to 1.42) | |
| Quadratic term | 1.03 (1.02 to 1.04) | | 0.94 (0.90 to 0.98) | |

*Sociodemographic variables adjusted for all sociodemographic variables in the table. Clinical variables adjusted for all variables in the table.
†Analysis of associations with clinical variables restricted to urban patients.
‡Weight is scaled (divided by 10) and centred on mean weight in the analysis.

**Table 3** Reported reasons for leaving care or changing clinics among 111 patients who were traced and found alive

| | No longer in care | | |
|---|---|---|---|
| **Reason for no longer attending clinic** | **Urban (n=41)** | **Rural (n=30)** | **All (n=71)** |
| Lack money for transport | 11 (26.8%) | 15 (50.0%) | 26 (36.6%) |
| Does not feel sick | 17 (41.5%) | 2 (6.7 %) | 19 (26.8%) |
| Travelled/moved away | 16 (39.0%) | 3 (10.0%) | 19 (26.8%) |
| Health centre is far away | 5 (12.2%) | 13 (43.3%) | 18 (25.4%) |
| Lack time | 8 (19.5%) | 3 (10.0%) | 11 (15.5%) |
| Gets cotrimoxazole from other sources | 7 (17.1%) | 3 (10.0%) | 10 (14.1%) |
| Doubts HIV status | 4 (9.8 %) | 4 (13.3%) | 8 (11.3%) |
| Fear of being seen at the HIV clinic | 0 (0.0 %) | 5 (16.7%) | 5 (7.0 %) |
| Does not like drugs/side effects | 4 (9.8 %) | 1 (3.3 %) | 5 (7.0 %) |
| Using herbal/traditional medicines | 0 (0.0 %) | 3 (10.0%) | 3 (4.2 %) |
| Other reason | 3 (7.3 %) | 2 (6.7 %) | 5 (7.0 %) |
| | In care at another clinic | | |
| **Reason for changing clinics** | **Urban (n=23)** | **Rural (n=17)** | **All (n=40)** |
| Closer to work | 12 (52.2%) | 6 (35.3%) | 18 (45.0%) |
| Lack of money for transport | 6 (26.1%) | 4 (23.5%) | 10 (25.0%) |
| Less waiting time | 7 (30.4%) | 0 (0.0 %) | 7 (17.5%) |
| Lack time | 5 (21.7%) | 1 (5.9 %) | 5 (12.5%) |
| Friends/family attend | 3 (13.0%) | 0 (0.0 %) | 3 (7.5 %) |
| Fear of being seen at the first clinic | 3 (13.0%) | 1 (5.9 %) | 3 (7.5 %) |
| Better service | 1 (4.3 %) | 1 (5.9 %) | 2 (5.0 %) |
| Other reason | 1 (4.3 %) | 4 (23.5%) | 5 (12.5%) |

from the clinic registers alone, loss from care was nearly 65% higher than after correcting for outcomes among individuals who were traced. We found that a third of the patients who were considered lost were continuing to access care at another clinic (silent transfers). We also identified deaths that had not been reported to the clinic. Other studies that have used a sampling-based approach to correct estimates of retention among HIV-positive individuals on ART have had similar findings.[12 16 17]

A study among HIV-positive people in pre-ART care at two large clinics in Uganda in 2008–2011 found that loss from care after 2.5 years was 30.5% but decreased to 11.8% after correcting for outcomes in a sample of lost patients.[18] These figures are much lower than we found in our study in 2015, particularly in the rural clinics where corrected estimates of loss from care after 9 months were still 28.8%. The tracing period in our study was shorter and our definition of loss from care was more restrictive (3 months late to appointment vs 6 months late). Furthermore, some of the clinics in our study were smaller and more rural, so factors such as lack of transport or distance to the clinic may have presented greater barriers to retention. In the rural areas, patients often have to travel more than 10 km on foot or bicycle to get to the clinics. Lastly, the CD4 threshold for ART eligibility in the earlier study was ≤350, versus ≤500 in 2015, so a larger proportion of patients in our study may

have been asymptomatic and thus less motivated to remain in care.

Our estimates of retention, even after correction, are in line with previous studies in SSA that have shown poor retention among patients in pre-ART care. A recent review found a median of 53% of patients who had linked to pre-ART care were retained until the study endpoint.[19] Even among patients who have been identified as ART eligible, a not insignificant proportion may be lost before starting ART. A study of ART-eligible patients at a clinic in Uganda found 20% did not start ART within a year, with 8% dying while waiting to initiate ART.[11] Two separate reviews of retention in HIV care in SSA found that around a third of patients who were eligible for ART were lost before starting treatment.[5 20] Factors associated with loss from care in this stage include facility-level barriers such as requirements for multiple clinic visits, inflexible clinic hours, lengthy waiting times and poor quality of care, and individual-level barriers such as fear of HIV disclosure, or limited understanding of HIV.[21]

With the new WHO 'treat all' guidelines, all individuals will be eligible to start ART immediately, but in practice there is likely to be a delay between linking to care after testing positive and initiating treatment. Removing the CD4 eligibility threshold may increase the number of patients attending the clinics, which can put a strain on already overburdened healthcare systems. Many of the same barriers to

ART initiation will remain under 'treat all' unless the process of starting ART is made more efficient. For successful implementation of the new guidelines, it will be essential to have accurate estimates of the proportion of people who disengage from care in the period before starting ART.

In our study, most of the reported reasons for leaving care were economic (lack of money for transport, distance from the clinic) or health systems factors (moving to a location without an HIV care facility). These factors have been commonly cited in other studies and are a challenge to providing lifelong HIV care in resource-limited settings. A systematic review of linkage to and retention in HIV care found that transport costs and distance were two of the main barriers to retention in pre-ART care.[21] A considerable number of patients reported obtaining cotrimoxazole from other sources, presumably in response to the challenges they faced attending the clinic. Psychological factors such as feeling well, or not believing that one was HIV positive, were also cited as reasons for leaving care, especially among urban patients. As has been reported in other studies, we found that younger patients were more likely to be lost from care.[21] These findings suggest that a combination of interventions may be required to improve retention in care.

We used a pragmatic approach to correct our estimates of loss from care, arithmetically upweighting the outcomes of patients who were tracked successfully to represent those of patients who were lost. Other methods have been proposed for incorporating these outcomes, including using regression models to estimate the inverse probability weights, and multiple imputation in conjunction with the ascertained outcomes. Simulations have shown that these strategies all provide less biased results than the standard uncorrected approach that is used in many epidemiological studies.[22]

Our study has several limitations. We traced only a sample of patients who were lost and were able to find 61% of those who were selected for tracing. The individuals that we found may not have been representative of all patients who were lost. Although there was no evidence that the characteristics of those who were successfully traced were different from those who were not found, our small sample size means that we may not have had power to detect true differences if they existed, and residual selection bias may still remain. In addition, our sample size was based on practical considerations, rather than the power to detect a particular effect size. Our analysis of predictors of loss from care was underpowered to detect anything except large effects, particularly in our analysis of clinical factors, which was restricted to the urban clinics. Similarly, our estimates of retention in care, and of mortality, are less precise than they would have been with a larger sample. Furthermore, there were relatively few deaths so our sample size may not have been adequate to obtain an accurate estimate of mortality. We looked at loss from care over a fairly short period (9 months); it is possible that some of the individuals defined as lost based on the clinic registers would have returned to the clinic at a later date. For the individuals who were successfully traced, we relied on self-report to define whether an individual was still in care

at another clinic, which may have led to over-reporting of care. Our analysis of factors associated with loss from care was limited by the small number of covariates and the large amount of missing data in the clinic databases particularly from the rural areas. Our findings from government clinics in Uganda may not be generalisable across all HIV treatment programmes in SSA, where reasons for disengagement from care, and outcomes after disengagement, may differ.

In summary, we found that estimates of loss from pre-ART care using a sampling-based approach were substantially lower than those based on the clinic registers alone. Retention was much lower in rural clinics than in urban clinics and was in line with previous reports of pre-ART retention in SSA. Structural factors were a key barrier to retention. These findings may have implications for the successful implementation of the 'treat all' guidelines and retention in care among individuals with high CD4 counts in similar resource-limited settings.

**Acknowledgements** We acknowledge the Kampala City Council Authority clinic staff and officials, the Infectious Disease Institute Outreach team and the HIV care centres in Hoima and Kabaale (Dwoli health centre III, Kigorobya health centre IV and Kagadi hospital) for their support and collaboration. We thank the study participants for their participation and the project staff for their work. We acknowledge the assistance of the Statistics unit at the Infectious Disease Institute.

**Contributors** PN, ANK, JB and KB conceived and designed the study. PN, ANK and AK conducted the study. PN analysed the data and developed the first draft. ANK, JB and KB advised on data analysis. All authors contributed to the interpretation of the data, revised the article critically and approved the final version.

**Funding** This work was supported by the European and Developing Countries Clinical Trials Partnership (EDCTP) through project MF.2013.40205.020; however, EDCTP cannot accept any responsibility for information or views expressed herein. JB and KB receive support from the MRC UK and DFID–MRC Grant Reference MR/K012126/1. This award is jointly funded by the UK Medical Research Council (MRC) and the UK Department for International Development (DFID) under the MRC/DFID Concordat agreement and is also part of the EDCTP2 programme supported by the European Union.

**Competing interests** PN and JB had research grant support from EDCTP for the submitted work; KB and JB receive research grants from MRC UK and DFID; no financial relationships with any organisations that might have an interest in the submitted work in the previous 3 years.

**Patient consent** Not required.

**Ethics approval** London School of Hygiene and Tropical Medicine Ethics Committee (Ref 10334), Makerere University School of Public Health Higher Degrees Research and Ethics Committee (Ref 353), Infectious Diseases Scientific Review Committee and the Uganda National Council for Science and Technology (Ref 3998).

**Provenance and peer review** Not commissioned; externally peer reviewed.

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
