## [Reviewer comments · BMJ Open]

ARTICLE DETAILS

TITLE (PROVISIONAL)	Correction of estimates of retention in care among a cohort of HIV-positive patients in Uganda in the period before starting ART: a sampling-based approach
AUTHORS	Nyakato, Patience; Kiragga, Agnes; Kambugu, Andrew; Bradley, John; Baisley, Kathy

VERSION 1 – REVIEW

REVIEWER	Kathrin Zürcher Institute of Social and Preventive Medicine (ISPM), University of Bern, Switzerland
REVIEW RETURNED	07-Jun-2017

GENERAL COMMENTS	Dear authors This is a nice paper on tracing of pre-ART patients just before the WHO released the new treatment guideline recommending that all HIV positive patients should initiate ART. With this new approach “test and treat” the question is how important is this paper among pre-ART patients since nowadays all should initiate ART. However, the reality is different. There are still many pre-ART patients waiting for initiation, not be traced and informed after the new guidelines were released or through ARV stock outs patients can not be initiated. Here a few comments or questions: Introduction: The introduction gives a nice overview in general. I think the introduction could be improved if the authors tell us a bit more about how the approach “test and treat” was implemented in Uganda. Page 4, line 31/32: Even among those who were ART eligible, only a median of 66% initiated ART. Please add Reference. Study design: The author said that he size of the sample was based on practical considerations of how many patients could be traced at each clinic, rather than on formal sample size. My questions regarding the practical consideration: - Was the duration of tracing pre-defined? - What is the average number of patients that could be traced by one tracer per day? Page 6, line 41-46: Tracing was done between November 2015– March 2016 for both the urban and rural clinics. Tracers attempted to contact patients through phone calls and home visits, using
--

	addresses and locator forms containing secondary phone numbers, areas of residence, and a map to the area of residence. - How frequent and how many times did you called and/or visited the patient lost to follow-up at home? Ethics: Maybe I misunderstand but did all the patients give an informed consent to be traced? And you asked all the successfully traced patients for addition consent to obtain the reasons? Results: Page 9, line 14-18: The mean (SD) age of all patients who registered in pre-ART care was 30.6 years (9.0); the majority (68.2%) were females, and the median (IQR) CD4 count was 645 (529–834) cells/uL. - Please either use the median or mean. Is confusing since you use the median in the abstract. Table 1: - Why do you have patient with a CD4 count below 500? Is it a baseline CD4 count? In the study design the authors wrote that a list of all individuals who tested HIV positive between January-May 2015 (Kampala clinics) and January-August 2015 (rural clinics) and had a CD4 >500 cells/uL was obtained from the routine clinic records. Please clarification! - Please mention the % of the missing CD4 counts. - How do you deal with the missing CD4 counts? Imputations? Table 2: - Could you include the weight in table 1? - How do you deal with the missing CD4 counts?
--	--

REVIEWER	Ilesh Jani Instituto Nacional de Saúde, Mozambique
REVIEW RETURNED	10-Jul-2017

GENERAL COMMENTS	The authors describe important work on pre-ART retention in a time when ART for every patient, independently of their immunity level, is being recommended. The paper is well written and presents interesting evidence on the role of "silent transfers" and early mortality on the overall pre-ART retention. I have three comments: 1) More details should be given about the clinics where the study took place. Some of the study finding may be specific to the characteristics of the clinics. 2) Since tracing plays a major role in the outcome measurement in this research, more details about how it was done are warranted in the Methods section. Who were the tracers? How was the information collected by tracers? Was a standardised questionnaire used by tracers? 3) Sample calculation was not done for the study. This fact and its possible effects on the results should be discussed as a limitation of the study (on page 15).
--

REVIEWER	Tony Fitzgerald University College Cork Ireland
REVIEW RETURNED	14-Nov-2017

GENERAL COMMENTS	Review of Correction of estimates of retention in care among a cohort of HIV-positive patients in Uganda in the period before starting ART – a sampling based approach I congratulate the authors on a well written and carefully constructed paper. This may not seem consistent with the comments that follow but I feel that the logic needs to be more clearly stated. I have a few queries that I hope the authors can address. Initially, 307(26.6%) of subjects were described as lost to care. A further analysis successfully traced 60.5% of a sample of these subjects. Of those traced, some had died (5.9%), some were ‘actively’ in care in other clinics (33.9%), and some were out of care (60.2%). Minor points  1. Can the authors clarify what is meant by ‘actively in care’ 2. I am a little confused as to exactly what ‘lost to care’ means. If someone moves clinic then they can be in care, at the new clinic, but could also be lost to care at the new clinic if they had missed an appointment at that clinic by more than 3 months. Is this correct? 3. For those who died, did the authors ascertain whether they were in care at the time of death? Is it possible for the clinic to learn of someone’s death even though they were at that time lost to care? So, someone misses an appointment by more than 3 months and subsequently dies. Should that person be considered lost to care because the lost to care proceeded death? 4. What is meant by ‘lower order polynomials’ (line 32, page 8) 5. For the results in table 2, it’s not clear whether increasing weight is associated with increased or decreased risk of loss to care. Clearly as the association is quadratic it’s not possible to answer directly but the authors could make an attempt. For example, in the uncorrected model the risk of loss to care falls with weight initially but increases after weight equal to etc. Unless weight is centred it would seem that the quadratic term will dominate. 6. Given that the number tracked at the various clinics were not proportional to size, was this allowed for in later analysis/estimation? Major points The Statistical analysis section (line 47, page 7) states that patients who were successfully traced were weighted using inverse probability weights equal to the inverse proportion of patients successfully traced to those lost. Patients who were not found, or not selected for tracing, were given a zero weight and those still in care were given a weight on
---

	one. 7. However, some of those successfully traced could be out of care e.g. its more than 3 months since they should have attended the clinic but you still trace them. It seems you only need to reduce the initial estimate of %lost to care so as to allow for those who were in fact still in care or had died while in care. As such, you should be using the proportion of those still in care (at another clinic) or dead while in care to those lost. Clearly I'm not following the methods used so ask that you clarify. Consider the following example. Suppose you have 100 patients and it appears that 30 are lost to care. You randomly select 20 of the 30 for tracing. For convenience I presume there are no deaths. You successfully trace 10 of these and find that 6 are still alive and in care at another clinic. The 'inverse proportion of patients successfully traced among all patients who were lost' = $30/10 = 3$. If you apply the weights as described in the paper you get $70 \times 1 + 10 \times 3 = 100$. Alternatively, if we set to zero the weight for those successfully traced but found to be out of care then we get: $70 \times 1 + 6 \times 3 = 88$. I think the correct adjusted estimate of number still in care is $70 + 30 \times (6/10) = 88$ where $6/10$ is the proportion of those apparently lost who were in fact still in care. This agrees with the method described by the authors provided you set to zero weights for those found to be out of care. 8. I could not find any mention of the sensitivity analysis in the Results.
--	--

VERSION 1 – AUTHOR RESPONSE

Dear Editor,

RE: Response to the reviewers' comments: "Correction of estimates of retention in care among a cohort of HIV-positive patients in Uganda in the period before starting ART-a sample based approach"

Thank you very much for considering our manuscript for review and providing us the opportunity to respond to the reviewers' comments. We thank the reviewers for their very helpful comments.

Please find below our detailed response to the each issues raised by the reviewers. Also, please find attached the revised manuscript.

We have tried to address all of the issues that were raised by the reviewers. As a result, our manuscript is now over the suggested 4000 word limit (4314 words). To reduce the length, we would be happy to move some of the additional information (e.g. details about the tracers, an example showing how the weights are applied) to a supplementary appendix, if possible.

Your kind consideration of our submission will be highly appreciated.

Yours faithfully,
Patience Nyakato
Corresponding author

Response to reviewer's comments
Reviewer 1

1) Introduction

The introduction gives a nice overview in general. I think the introduction could be improved if the authors tell us a bit more about how the approach "test and treat" was implemented in Uganda.

Response: Uganda officially rolled out the test and treat guidelines in November 2016. By end 2017, nearly all government facilities had implemented test and treat. However, in practice, priority for ART initiation is given to existing patients in pre-ART care. Furthermore, ART 'stock outs' are still common, so individuals who are newly diagnosed are likely to have some period of pre-ART care. In March 2017, an estimated 6% of HIV positive persons who were enrolled in care were not on ART [1]. In addition, an estimated 48% of men who had tested HIV positive have not yet initiated ART [1]. We have added this information to the Introduction.

Page 4, line 31/32: Even among those who were ART eligible, only a median of 66% initiated ART. Please add Reference.

Response: Thank you for pointing this out. The reference is the same as on the previous line. We have inserted the reference on line 31/32 to clarify.

2) Study design

The author said that the size of the sample was based on practical considerations of how many patients could be traced at each clinic, rather than on formal sample size. My questions regarding the practical consideration:

Was the duration of tracing pre-defined?

Response: No, the duration of the tracing period was not pre-defined. The number of patients that could be traced was determined primarily by budgetary considerations, and the resources that were available to employ tracers and to intensively trace patients (transport costs, etc).

What is the average number of patients that could be traced by one tracer per day?

Response: In the urban clinics, the average number of patients that could be traced was about 2–3 per day. This was facilitated by the availability of working phone numbers, and accurate information provided on the locator forms, as well as the proximity of the patients to the clinics and locations on paved roads, so that home visits did not require long travel times. In the rural clinics, however, the average number of patients that could be traced was around 1 per day. Most of the rural patients did not have working phone numbers and the locator forms were never filled as accurately as in the urban clinics, so finding the patients was much more challenging. Most of the rural patients lived in areas where there were few paved roads, so the tracing of patients at their homes was done on bicycle or on foot. Please see response to Reviewer 2, point 2, for more information about the tracers, which has also been added to the Methods section.

Page 6, line 41-46: Tracing was done between November 2015–March 2016 for both the urban and rural clinics. Tracers attempted to contact patients through phone calls and home visits, using

addresses and locator forms containing secondary phone numbers, areas of residence, and a map to the area of residence.

How frequent and how many times did you called and/or visited the patient lost to follow-up at home?

Response: An initial round of phone calls was made to establish contact with the patients, explain the reason for the call and ask them for consent to be interviewed. Patients were given an appointment date for an interview at their home, the clinic or a place of their choice. A second round of phone calls was then done to remind patients of their appointments the day before the visit. For those whose phone numbers were unreachable at first attempt, two further phone call attempts were made, as well as at least three attempts to secondary phone contacts. Phone calls were made over 3–4 days, at different times of the day. For those who could not be reached by phone, the tracers then attempted to locate the patients at home, using locator forms. Tracing was often done over several days because the information on the locator forms was inaccurate, especially in the rural areas, or the patient was found to have moved to a new location in the area. For patients who had agreed to attend an interview at the clinic but did not make it to their interview appointment, a home visit was offered as an alternative to make it more convenient for them. We have added further information about the tracing to the Methods section.

3) Ethics

Maybe I misunderstand but did all the patients give an informed consent to be traced? And you asked all the successfully traced patients for addition consent to obtain the reasons?

Response: When patients register at the clinic as part of routine HIV care, they are asked for consent to be traced in case they miss their appointments. It is at this point that they provide their phone number and secondary contact information, and fill out a locator form. We used this contact information provided to the clinics to trace patients. All patients who were successfully traced were informed about our study, and asked for consent to be interviewed. This has been clarified in the Ethics section.

4) Results

Page 9, line 14-18: The mean (SD) age of all patients who registered in pre-ART care was 30.6 years (9.0); the majority (68.2%) were females, and the median (IQR) CD4 count was 645 (529–834) cells/uL.

Please either use the median or mean. Is confusing since you use the median in the abstract.

Response: We are sorry for the confusion. We have changed to the median throughout.

Table 1:

Why do you have patient with a CD4 count below 500? Is it a baseline CD4 count? In the study design the authors wrote that a list of all individuals who tested HIV positive between January-May 2015 (Kampala clinics) and January-August 2015 (rural clinics) and had a CD4 >500 cells/uL was obtained from the routine clinic records. Please clarification!

Response: At the time of the study, Uganda had officially implemented the CD4 <500 threshold for ART initiation. However, priority was given to individuals with CD4 <350, so in some clinics, individuals with CD4 between 350 and 500 were still in the pre-ART care programme. This clarification has been added to the Methods section page 6 (Study Setting sub-section).

Please mention the % of the missing CD4 counts.

Response: The missing CD4 counts are described in Table 1; this information has also been added to the Results section on page 10.

How do you deal with the missing CD4 counts? Imputations?

Response: All patients with missing CD4 counts were from the rural clinics. Information on other clinical covariates (e.g. weight, WHO stage) was also missing for many of the rural patients, because of the quality of the record keeping. When comparing characteristics between the patients who were in care and lost, or who were successfully traced or not (Chi-squared test p-values in Table 1), the missing data are excluded from the comparison (this information is in the table footnote). Because of the high proportion of missing clinical data in the rural clinics (74% of all rural patients, including those still in care), we did not attempt to impute these values.

Table 2:

Could you include the weight in table 1?

Response: We have added weight to Table 1.

How do you deal with the missing CD4 counts?

Response: As described above, 74% of patients in the rural clinics were missing CD4 counts, and other clinical data. Because of this high proportion of missing data, the analysis of factors associated with loss from care in Table 2 was restricted to the urban clinics, where data were essentially complete. This is stated in the Statistical Methods section (page 9), and in the Table 2 footnote.

Reviewer 2

1) More details should be given about the clinics where the study took place. Some of the study finding may be specific to the characteristics of the clinics.

Response. The study took place at 3 urban health centres in Kampala (Kisenyi, Kawala and Kitebi) which are under the Kampala City Council Authority, and 3 rural facilities in western Uganda which are run by the Uganda Ministry of Health. All 6 facilities are supported by the Infectious Disease Institute, Makerere University. The urban facilities are all level IV health centres, which have a target catchment population of 100,000. Kisenyi is the largest of these, and serves a particularly economically impoverished area. The rural facilities included one level IV (Kiogorobya) and one level III (Dwoli) health centre in Hoima district, and Kagadi District Hospital, in Kibaale district. Hoima district had an estimated population of 574,000 in the 2014 census, and Kibaale had an estimated population of 789,000. This information has been added to the methods section.

2) Since tracing plays a major role in the outcome measurement in this research, more details about how it was done are warranted in the Methods section. Who were the tracers? How was the information collected by tracers? Was a standardised questionnaire used by tracers?

Response: We employed trained nurse counsellors for the patient interviews. All of the nurse counsellors had previous research experience at the Infectious Disease Institute, and the interviews all used a structured questionnaire. The nurse counsellors made the contact attempts by telephone (please see response to Reviewer 1, point 3 under Study Design) and, if the locator information was sufficient (mostly patients from urban clinics), traced patients in the community who could not be contacted by phone. For urban patients who could not be located from the forms, and for all rural patients who could not be contacted by phone, we used expert patients from the clinics as tracers,

paired with the nurse counsellors. Expert patients serve as community volunteers at the clinic, and offer support to fellow patients in HIV care. They are familiar with the surrounding community and are often called on by the clinic to help trace patients who have missed their visits. We have added more information about the tracers to the Methods section.

3) Sample calculation was not done for the study. This fact and its possible effects on the results should be discussed as a limitation of the study; -page 15.

Response: The sample size for this study was not based on the power to demonstrate a particular effect size, and we agree with the reviewers that this is a potential limitation of our analyses of factors associated with loss from care. We have added this to the limitations in the Discussion (page 18).

Reviewer 3

Initially, 307(26.6%) of subjects were described as lost to care. A further analysis successfully traced 60.5% of a sample of these subjects. Of those traced, some had died (5.9%), some were 'actively' in care in other clinics (33.9%), and some were out of care (60.2%).

Minor points

1. Can the authors clarify what is meant by 'actively in care'?

Response: By 'actively in care', we mean that the patients provided proof of registration at other clinics through patient cards that had been issued to them, with evidence of having attended recent appointments on these cards. However, for patients who were interviewed on the phone (N=20 of the 118 who were located), we had to rely on self-report of still being in care. We asked the patients the name of the clinic where they were registered, and the date of their next appointment, and considered them to still be in care if they could give this information. We have added these details to the Methods section.

2. I am a little confused as to exactly what 'lost to care' means. If someone moves clinic then they can be in care, at the new clinic, but could also be lost to care at the new clinic if they had missed an appointment at that clinic by more than 3 months. Is this correct?

Response: Please see our response to point 1 above. We used patient clinic cards (for those who were interviewed in person) or self-report of the next appointment date to assess whether the patient was in care at another clinic. Patients who had transferred to other clinics but did not have a current appointment date were not considered to be in care.

3. For those who died, did the authors ascertain whether they were in care at the time of death?

Response: We were not able to ascertain if the patients were still in care at the point of death, except for one patient whose family member reported that the individual was still in care, but had died in an accident. Please see our response below for additional clarification.

Is it possible for the clinic to learn of someone's death even though they were at that time lost to care?

Response: Yes, it is possible for the clinic to learn about a patient's death through the clinic's routine tracing programme. These programmes have been intensified over the past years; however, they are primarily targeted towards patients who are on ART and there is less effort placed on tracing pre-ART patients. However, anecdotally, there are also instances of clinic staff being aware of a patient's

death, e.g. if reported by a family member, but this information is not always recorded in the clinic registers.

So, someone misses an appointment by more than 3 months and subsequently dies. Should that person be considered lost to care because the lost to care preceded death?

Response: We used a competing risk analysis to take deaths into account. In this analysis, individuals whose date of death is more than 3 months after their last scheduled appointment would be considered as lost, for the reasons that you have described. Individuals whose date of death was within 3 months of their last scheduled appointment would not be considered lost, and instead would be considered to have died when they were still in care. We have clarified in the Methods section that it is the earliest of these dates that is used in the analysis.

4. What is meant by 'lower order polynomials' (line 32, page 8)

Response: By 'lower order polynomials', we mean quadratic or cubic power terms. We have clarified this in the Methods section.

5. For the results in table 2, it's not clear whether increasing weight is associated with increased or decreased risk of loss to care. Clearly as the association is quadratic it's not possible to answer directly but the authors could make an attempt. For example, in the uncorrected model the risk of loss to care falls with weight initially but increases after weight equal to etc. Unless weight is centred it would seem that the quadratic term will dominate.

Response: In the uncorrected analysis, the risk of loss to follow-up decreased with increasing weight to around 60 kg, and then increased. In the corrected analysis, the relationship with weight changed, with the risk of loss to follow-up increasing slightly with increasing weight to around 60 kg and then decreasing. Weight has been centred and scaled (divided by 10) in the analysis. This information has been added to the Results section and as a footnote to Table 2.

6. Given that the number tracked at the various clinics were not proportional to size, was this allowed for in later analysis/estimation?

Response: The weights were estimated separately for each clinic, so take into account the proportion of lost patients that were sampled in each clinic.

Major points

The Statistical analysis section (line 47, page 7) states that patients who were successfully traced were weighted using inverse probability weights equal to the inverse proportion of patients successfully traced to those lost. Patients who were not found, or not selected for tracing, were given a zero weight and those still in care were given a weight on one.

7. However, some of those successfully traced could be out of care e.g. its more than 3 months since they should have attended the clinic but you still trace them. It seems you only need to reduce the initial estimate of %lost to care so as to allow for those who were in fact still in care or had died while in care. As such, you should be using the proportion of those still in care (at another clinic) or dead while in care to those lost. Clearly I'm not following the methods used so ask that you clarify.

Consider the following example. Suppose you have 100 patients and it appears that 30 are lost to care. You randomly select 20 of the 30 for tracing. For convenience I presume there are no deaths. You successfully trace 10 of these and find that 6 are still alive and in care at another clinic. The

'inverse proportion of patients successfully traced among all patients who were lost' = $30/10 = 3$. If you apply the weights as described in the paper you get $70*1 + 10*3 = 100$.

Alternatively, if we set to zero the weight for those successfully traced but found to be out of care then we get: $70*1 + 6*3 = 88$.

I think the correct adjusted estimate of number still in care is $70 + 30*(6/10) = 88$ where $6/10$ is the proportion of those apparently lost who were in fact still in care. This agrees with the method described by the authors provided you set to zero weights for those found to be out of care.

Response. For our method, using your example above, this is how we computed the weights:

70 in care * weight 1,
10 not tracked *weight 0
10 tracked but not found*weight 0,
10 tracked successfully: 6 in care*weight 3, 4 out of care*weight 3

So $(70*1) + (6*3) + (4*3) = 100$ (the weights must sum to 100 since they are making the individuals who are still in care and the ones who were successfully traced to be equal to the original patient population of $N=100$)

The number in care would then be $(70*1) + (6*3) = 88$ so total the total proportion in care is $88/100$ which is the same figure you arrived at in your calculation.

We have attempted to clarify this further in the Methods by including this example.

8. I could not find any mention of the sensitivity analysis in the Results.

Response: The results of the sensitivity analysis is mentioned in the third paragraph of the Results section (page 9)

References

1. Uganda AIDS Commission. Presidential Fast Track Initiative on Ending HIV and AIDS in Uganda. June 2017.
<http://www.aidsuganda.org/images/documents/PresidentialHandbook.pdf>

VERSION 2 – REVIEW

REVIEWER	Kathrin Zürcher Institute of Social and Preventive Medicine (ISPM), University of Bern, Switzerland
REVIEW RETURNED	25-Dec-2017

GENERAL COMMENTS	Well done. Thank you very much for answering all the questions.
---

REVIEWER	Ilesh Jani Instituto Nacional de Saúde, Mozambique
-----------------	---

REVIEW RETURNED	05-Dec-2017
GENERAL COMMENTS	Most of my comments on the initial submission were responded by the authors. However, I still feel that the authors should include some discussion on the possible effect of "sampling based on practical considerations". Currently, the authors have only included one sentence in the Discussion stating that this is a limitation, but some explanation on how it possibly affects the results should be included.
REVIEWER	Tony Fitzgerald University College Cork Ireland
REVIEW RETURNED	08-Jan-2018
GENERAL COMMENTS	They authors have addressed all my concerns with one exception. I recommend the paper be accepted but ask that the authors address the following. The authors in their reply (point 6) state that the weights were calculated for each clinic separately. I understand how and why this was done. However, before aggregating the results across urban and rural clinics to obtain overall estimates, such as those given in Table 3 and in the text, they still need to allow for the difference between the %breakdown by clinic in the sample and the %breakdown in the population. For any outcome/characteristic that varies between urban and rural centres, the estimates based on the combined sample will depend on the proportion of the sample chosen from urban as opposed to rural areas. From the first paragraph of the results, "the total number lost to care was 307, 207 from urban clinics and 100 from rural clinics" (2:1). However the numbers selected for tracing was 116 and 79 respectively (3:2). So the rural clinics are over-represented. For example, it appears from table 3 that the reasons for no longer attending clinic differ between urban and rural centres. The separate results for urban and rural are OK but the results for 'All' reflect the sample proportion in urban/rural and are a biased estimate of the population figure. The authors do a good job of presenting results separately for urban and rural centres so this not a major concern. Final comment: Motivated by a question I asked the authors added a few sentences to the Statistical Analysis (page 9) For example, in a clinic with 100 patients of whom 30 were lost, 10 were successfully traced and 6 were found to be still in care, the weights for the patients who were traced would be $30/10=3$. The corrected estimate for the proportion in care would be calculated as: $[70 \times 1 + 6 \text{ (found to be still in care)} \times 3] / [70 \times 1 + 6 \times 3 + 4 \text{ (found to be out of care)} \times 3] = 88/100.$

VERSION 2 – AUTHOR RESPONSE

17 January 2018

The Executive Editor,
BMJ Open

Dear Editor,

RE: Response to the second set of reviewers' comments: "Correction of estimates of retention in care among a cohort of HIV-positive patients in Uganda in the period before starting ART-a sample based approach"

I thank you once again for considering our manuscript and for the reviewer's helpful comments.

Please find below our considered response to the remaining minor comments.

Yours faithfully,
Patience Nyakato
Corresponding author

Response to reviewer's comments

Reviewer 1

Well done. Thank you very much for answering all the questions.

Response: Thank you so much for your very helpful comments and we are glad that we addressed all your questions

Reviewer 2

Most of my comments on the initial submission were responded by the authors. However, I still feel that the authors should include some discussion on the possible effect of "sampling based on practical considerations". Currently, the authors have only included one sentence in the Discussion stating that this is a limitation, but some explanation on how it possibly affects the results should be included.

Response: Thank you for this comment. We have expanded our discussion on the effect of our sample size on page 18, as follows: Our analysis of predictors of loss from care was underpowered to detect anything except large effects, particularly in our analysis of clinical factors which was restricted to the urban clinics. Similarly, our estimates of retention in care, and of mortality, are less precise than they would have been with a larger sample size. Furthermore, there were relatively few deaths so our small sample size may not have been adequate to obtain an accurate estimate of mortality.

Reviewer 3

The authors have addressed all my concerns with one exception. I recommend the paper be accepted but ask that the authors address the following.

The authors in their reply (point 6) state that the weights were calculated for each clinic separately. I understand how and why this was done. However, before aggregating the results across urban and rural clinics to obtain overall estimates, such as those given in Table 3 and in the text, they still need to allow for the difference between the %breakdown by clinic in the sample and the %breakdown in the population.

For any outcome/characteristic that varies between urban and rural centres, the estimates based on the combined sample will depend on the proportion of the sample chosen from urban as opposed to rural areas. From the first paragraph of the results, "the total number lost to care was 307, 207 from

urban clinics and 100 from rural clinics" (2:1). However, the numbers selected for tracing was 116 and 79 respectively (3:2). So the rural clinics are over-represented.

For example, it appears from table 3 that the reasons for no longer attending clinic differ between urban and rural centres. The separate results for urban and rural are OK but the results for 'All' reflect the sample proportion in urban/rural and are a biased estimate of the population figure. The authors do a good job of presenting results separately for urban and rural centres so this not a major concern.

Response: Thank you for these comments. The weighting in our method does take into account the different sampling fractions for the rural and urban clinics. The combined analysis includes all patients who registered for pre-ART care during the time of the study, rather than a sub-sample of the patient population. The weights serve to make the patients who were tracked at each clinic represent the outcomes in all patients who were lost at the clinic – therefore, in the combined analysis, the distribution of rural and urban patients remains exactly the same as in the original population. We would like to provide another example to illustrate what we did:

Suppose that there are 400 patients who register for pre-ART care in the urban clinics and 100 who register in the rural clinics (similar to the distribution in our study, with 925 in urban clinics and 228 in rural), and that N=80 are lost from the urban clinics and N=40 from the rural.

We then select N=40 to trace from urban and N=30 from rural – 50% of urban patients who were lost and 75% of the rural patients who were lost, so rural patients are over-represented in this sample of patients that we trace.

We then find N=20 from the urban clinics and N=20 from the rural (50% of the urban patients who were traced and 67% of the rural – so rural patients are also over-represented in the total sample of patients that we find). For simplicity, let's assume that all patients who were found were still in care at another clinic.

The weights for the urban patients who were successfully traced would be $80/20 = 4.0$, and for the rural patients would be $40/20 = 2.0$. We apply those weights within each rural/urban group to get:

Urban: $(320 \text{ still in care} * 1) + (40 \text{ not traced} * 0) + (20 \text{ traced but not found} * 0) + (20 \text{ found still in care} * 4) = 400$ - i.e. the original number in the urban clinics

Rural: $(60 \text{ still in care} * 1) + (10 \text{ not traced} * 0) + (10 \text{ traced but not found} * 0) + (20 \text{ found still in care} * 2) = 100$ - i.e. the original number in the rural clinics

So the combined analysis includes all the patients who registered for care in the clinics and the distribution after applying the weights will be the same as in the original population.

We hope that this example clarifies our methods.

Final comment:

Motivated by a question I asked the authors added a few sentences to the Statistical Analysis (page 9)

For example, in a clinic with 100 patients of whom 30 were lost, 10 were successfully traced and 6 were found to be still in care, the weights for the patients who were traced would be $30/10=3$. The corrected estimate for the proportion in care would be calculated as: $[70 \times 1 + 6 \text{ (found to be still in care)} \times 3] / [70 \times 1 + 6 \times 3 + 4 \text{ (found to be out of care)} \times 3] = 88/100$.

I asked that the author modify this to read

For example, suppose that in a clinic with 100 patients

So that it's clear the example is for illustration and is not referring to a real clinic included in your analysis.

Response: We have made this suggested change.

VERSION 3 – REVIEW

REVIEWER	Ilesh Jani Instituto Nacional de Saúde Mozambique
REVIEW RETURNED	06-Feb-2018
GENERAL COMMENTS	The authors have addressed all my comments.